# Spotlight: Mobile UI Understanding using Vision-Language Models with a Focus

**Gang Li**
Google Research
Mountain View, CA
`leebird@google.com`

**Yang Li**
Google Research
Mountain View, CA
`liyang@google.com`

## Abstract

Mobile UI understanding is important for enabling various interaction tasks such as UI automation and accessibility. Previous mobile UI modeling often depends on the view hierarchy information of a screen, which directly provides the structural data of the UI, with the hope to bypass challenging tasks of visual modeling from screen pixels. However, view hierarchies are not always available, and are often corrupted with missing object descriptions or misaligned structure information. As a result, despite the use of view hierarchies could offer short-term gains, it may ultimately hinder the applicability and performance of the model. In this paper, we propose *Spotlight*, a vision-only approach for mobile UI understanding. Specifically, we enhance a vision-language model that only takes the screenshot of the UI and a region of interest on the screen—the *focus*—as the input. This general architecture of Spotlight is easily scalable and capable of performing a range of UI modeling tasks. Our experiments show that our model establishes SoTA results on several representative UI tasks and outperforms previous methods that use both screenshots and view hierarchies as inputs. Furthermore, we explore multi-task learning and few-shot prompting capacities of the proposed models, demonstrating promising results in the multi-task learning direction.

## 1 Introduction

Computational understanding of mobile user interfaces (UI) is a crucial step for achieving intelligent UI behaviors such as UI automation, and addressing diverse interaction scenarios such as those requiring accessibility features. Recently, mobile UI understanding has attracted numerous research interests. Previous works have proposed various UI modeling tasks and datasets, including widget captioning (Li et al., 2020b), screen summarization (Wang et al., 2021), command grounding (Li et al., 2020a; Bai et al., 2021; Burns et al., 2022) and other tasks (Li et al., 2022; He et al., 2020) on the mobile screen. Many of these works focus on bridging natural language and graphical user interfaces, which have shown potential for enabling language-based interaction.

A mobile UI screen can come with a view hierarchy—a structural representation of the screen—in addition to the screenshot image. Using view hierarchy as input allows a model to directly acquire detailed information of UI objects such as their types, text content and positions on the screen, bypassing challenging visual modeling tasks such as inferring object information from screenshots (Li et al., 2021; Zhang et al., 2021). Previous works have shown the benefit of using view hierarchy in UI modeling in several tasks. For example, models using view hierarchy have achieved better performance than their vision-only counterparts in UI captioning tasks (Li et al., 2020b; Wang et al., 2021).

However, recent work has revealed that mobile UI view hierarchies often contain inaccurate information about the UI screen, e.g., missing object text and misaligned structure information. Li et al. (2022) showed that about 37.4% of the screen view hierarchies contain objects with invalid bounding boxes. Ross et al. (2018) showed that 92.0% of Floating Action Buttons had missing text labels, compared to 54.7% of Image Buttons and 86.3% of Clickable Images. These object text labels (e.g., `content_desc`) are among the most important features in view hierarchies. Removing text features resulted in a drop by 17 CiDER points for the widget captioning task (Li et al., 2020b).

Therefore, such inaccurate information in the input can seriously hinder models in realizing their full potential in UI modeling. Although recent work has proposed methods for repairing view hierarchies (Li et al., 2022), substantial effort is still needed to robustly denoise raw view hierarchies. On top of these, UI screen data does not always have view hierarchies available in them, such as mobile UI images crawled from the web. Fetching view hierarchy at runtime in a mobile environment also imposes additional system constraints for the applicability of models that rely on view hierarchies.

In this paper, we investigate the direction of using only visual UI screenshots as input (i.e., without including view hierarchies) for UI modeling tasks. We observe that many UI modeling tasks essentially aim to learn a mapping between the UI objects and text. As a result, vision-language models, a class of models that encode visual (and language) modalities and decode text answers, become a natural choice for the model architecture. Although previous works show that vision-only models generally perform worse than the models using both visual and view hierarchy input (Li et al., 2020b; Wang et al., 2021), we believe that visual language models offer two unique opportunities: 1) the simple architecture enables a model easily scalable, and 2) many heterogeneous tasks can be universally represented by the two core modalities of vision and language. These advantages have been evidenced by the recent successes of the vision-language models (Chen et al., 2022; Alayrac et al., 2022; Yu et al., 2022; Wang et al., 2022).

In contrast to previous visual-language tasks in the general domain, which usually use an entire image as input, UI modeling tasks are often concerned with a specific object or area on the screen. This requires a vision-language model to be able to focus on the object or area of interest. Thus, we propose *Spotlight*[1], which enhances a vision-language model to generate text responses with respect to a focus object or region to support various UI modeling tasks (see Figure 1). In our experiments, we initialize Spotlight by leveraging pretrained large ViT (Dosovitskiy et al., 2021) and T5 (Raffel et al., 2019) checkpoints. We then pretrain Spotlight with unlabeled datasets consisting of about 2.5 million mobile UI screens and 80 million web pages, which is followed by one of the three modeling strategies: single-task finetuning, multi-task finetuning or few-shot learning.

Our main contribution is three-fold. First, we propose a novel vision-language model architecture that is capable of finetuning, multi-task learning and few-shot learning for mobile UI tasks. The model can easily scale and generalize to other tasks without architectural changes. This model advances the art of UI understanding without needing to use view hierarchies as inputs that has many drawbacks in practice. Secondly, we develop a method for creating large-scale pretraining datasets from automatically collected mobile screens and web pages. These pretraining datasets and methods are crucial for our vision-language model to learn the prior knowledge of the unique domain of mobile screens and UIs. Finally, we conduct extensive experiments over the proposed model, including using various focus region representations and modeling strategies. Our experiments show that the proposed models obtain new SoTA performance in both single-task and multi-task finetuning for the four tasks, including widget captioning, screen summarization, command grounding and tappability prediction. We also examine the feasibility of using the proposed model for few-shot prompting.

## 2 RELATED WORK

UI modeling problems have drawn widespread interest from researchers in both the ML and HCI fields (Li et al., 2020b; Wang et al., 2021; Li et al., 2020a; Burns et al., 2022; Bai et al., 2021; Zhang et al., 2021; Wu et al., 2021; He et al., 2020). With the overarching goal of enabling intelligent UIs and addressing mobile accessibility, previous works have proposed a rich set of mobile UI modeling tasks, along with datasets and benchmarks. Widget captioning (Li et al., 2020b) aims to generate natural language description for UI objects on the screen. The capability can enable accessibility features such as the TalkBack[2] screen reader to improve user experience for vision-impaired users. Screen2Words expands UI captioning by proposing the task for summarizing the entire screen (Wang et al., 2021). Command grounding maps a natural language command to a UI object on the screen, via single or multi-step interactions (Li et al., 2020a; Burns et al., 2022; Bai et al., 2021). Tappability prediction predicts whether a UI object is tappable when perceived by human (Swearngin & Li, 2019; Schoop et al., 2022), which is useful for UI design validation. Most of these previous works

---

[1]The name draws the analogy that a spotlight illuminates a target region.
[2]https://support.google.com/accessibility/android/answer/6283677?hl=en

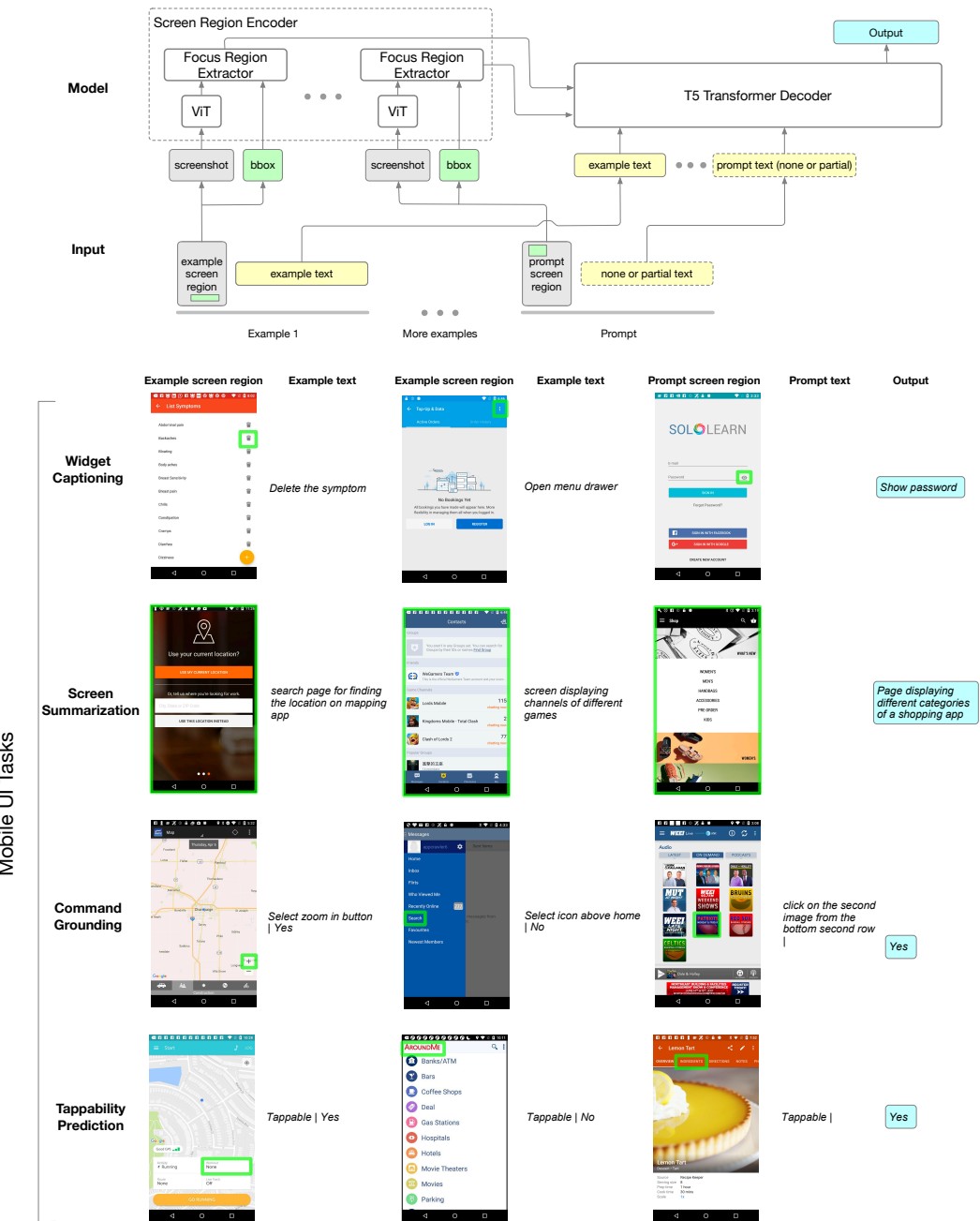

Figure 1: An illustration of the model architecture and UI task examples. ViT (Dosovitskiy et al., 2021) and T5 (Raffel et al., 2019) are initialized with checkpoints pretrained in the general domain of natural image and language.

used both the screenshot and the view hierarchy of a screen as input in their models, and revealed that multimodal input achieves better performance than vision-only variants (Li et al., 2020b; Wang et al., 2021). Prior works have also borrowed language modeling techniques such as BERT (Devlin et al., 2019) for representation learning to facilitate downstream tasks (Bai et al., 2021; He et al., 2020). Recently, Li et al. (2021) investigated the feasibility of multi-task modeling for achieving a range of UI tasks simultaneously. To combat the lack of accurate view hierarchy, prior works have investigated model-based approaches for repairing noisy view hierarchies (Li et al., 2022), or UI object detection over screenshots to derive information such as object attributes and structure

information (Zhang et al., 2021; Wu et al., 2021). In this work, we investigate vision-input-only approaches for several representative UI modeling tasks that have been established by prior works, including widget captioning, screen summarization, command grounding and tappability prediction.

Many of these mobile UI tasks are concerned with specific objects or areas on the UI screen. Previous works have employed different methods to allow a model to focus on an object or a region of interest on the screen. Li et al. (2020b); Wang et al. (2021); Bai et al. (2021) used the screenshot pixels cropped for individual objects as input to the model. Li et al. (2021) used a focus map to bias the ResNet outputs to pay more attention to the pixels corresponding to an object. Schoop et al. (2022) added a fourth channel to an RGB screenshot image that indicates the target region with a binary mask where the 1s correspond to the area of the object. The cropping and bias-based approach might result in losing the context of the object, despite providing direct localization of the object. While adding a fourth channel also keeps the information of the entire image, it is incompatible with existing vision models that are based on 3-channel RGB images, which makes it difficult to leverage their pretrained checkpoints. In this work, we investigate a variety of methods for region representations.

Recently, rapid progress has been seen for the large vision and language models, such as GPT-3 (Brown et al., 2020), T5 (Raffel et al., 2019), PaLM (Chowdhery et al., 2022), CLIP (Radford et al., 2021), DALL-E (Ramesh et al., 2022) and Imagen (Saharia et al., 2022). These models exploited large unlabeled data crawled from the web and demonstrate impressive finetuning and few-shot learning capacities. Vision-language models that encode images and decode text have leveraged large pretrained vision and language components and observed significant performance gain, e.g., Flamingo (Alayrac et al., 2022) and PaLI (Chen et al., 2022). Specifically, Flamingo showed that the few-shot prompting ability of the large language model can generalize to multimodal cases using multiple pairs of image and text as input. We adapt this model architecture to explore task-specific finetuning and multi-task learning as well as few-shot capability for mobile UI tasks.

## 3 Data

### 3.1 Pretraining Datasets

Two kinds of UI data are used to pretrain Spotlight models. First, we use the publicly available C4 corpus (Raffel et al., 2019) which contains a large amount of web pages that can be rendered into screenshots, similar to mobile UI screenshots. We use 80 million web page screenshots for pretraining. Secondly, we use a large-scale UI corpus that consists of 2.69 million mobile screenshots with paired view hierarchies. The dataset was collected by crawling various mobile apps. Screenshots and their view hierarchies are captured after performing random clicks in the Android emulator. Although being much smaller than the C4 dataset, the mobile data is critical as they are in-domain unlabeled data of our downstream tasks. As mentioned in Section 1, many UI modeling tasks aim to learn a mapping between a UI object and a language description. Therefore, we propose to pretrain Spotlight models to auto-regressively decode text-based attributes of individual objects on the web or mobile UI, with only the screenshot image and the object bounding box as the input.

**Web Page Data**: We select a list of HTML attributes as the text descriptions of an object based on a manual examination of their qualities. The selected attributes contain text either corresponding to the rendered text on the screen (e.g., a button label), or describing the functionality of the HTML object (e.g., `alt-text`). Text consisting only of generic words (e.g., *"image", "text"*) are removed. We also use the `title` of a web page as the description of the full screenshot. The statistics of the most common attributes with more than $10M$ instances are summarized in Table 1. The comprehensive statistics and list of generic words are reported in Appendix D.

| Web Pages | Objects | text | alt-text | aria-label | title | placeholder |
|---|---|---|---|---|---|---|
| 80M | 2.6B | 2.3B | 136.1M | 90.4M | 80M | 33.1M |
| Avg word # | | 5.1 | 3.8 | 2.7 | 10.7 | 2.2 |

Table 1: Statistics of the C4 pretraining dataset.

**Mobile Data**: Similar to prior work (Li et al., 2020b), we only use leaf objects as they are more likely to represent the functionalities of the UI. For each object, we use three text-based attributes from a view hierarchy node: `text`, `content_description`, and `resource_id`. These attributes are showed to be important features for models using view hierarchy as inputs (Li et al., 2020b; Wang et al., 2021). Note that we derive the output labels from view hierarchies but do not use the view hierarchies themselves as input.

To compensate for missing text in view hierarchies, we further run OCR on the screenshot to extract additional text that might be present on the screen. Recognized single-line text with 80% or higher confidence are used. To improve the data quality, we preprocess the dataset to remove URLs, Unicode, single-character and non-alphabetical strings. We also remove text that rarely occurs, i.e., fewer than 5 times in the entire dataset, or consists of generic phrases (e.g., *"text view"*) that do not convey useful information about the object. Table 2 summarizes the statistics of the mobile dataset.

| Screen/View Hiearchy | Objects | Text | Content Desc | Resource id | OCR Text |
|:---:|:---:|:---:|:---:|:---:|:---:|
| 2.5M | 51.6M | 12.1M | 2.2M | 13.9M | 29.1M |
| Avg word # | | 2.6 | 2.0 | 2.1 | 3.0 |

Table 2: Statistics of the mobile pretraining dataset.

## 3.2 UI TASK DATASETS

We select four representative UI modeling tasks that previous work proposed as downstream tasks (see Figure 1). We list these tasks below.

- Widget Captioning (Li et al., 2020b): Generating a natural language description for the functionality of a given object on the screen.
- Screen Summarization (Wang et al., 2021): Generating a high-level summary of a UI screen to describe its contents and functionalities.
- Command Grounding (Li et al., 2021): Finding the UI object on the screen that matches the intent of a natural language command.
- Tappability Prediction (Schoop et al., 2022): Predicting whether a given object on the screen is tappable or not.

These four tasks require models to learn the mapping between a UI object and a natural language description in either $object \rightarrow language$ or $language \rightarrow object$ direction. Furthermore, it is essential for the model to be able to both zoom in to focus on a specific region for tasks such as widget captioning, and zoom out to grasp the full screen for tasks such as screen summarization. We discuss these tasks in detail in Appendix E. In our experiments, we use the same dataset splits for comparison with these benchmarks.

## 4 MODELS

We design Spotlight based on the general encoder-decoder architecture that has been widely used for vision-language modeling (Chen et al., 2022; Alayrac et al., 2022; Wang et al., 2022). We employ ViT (Dosovitskiy et al., 2021) for encoding images (screenshots) and a Transformer decoder for generating language. In contrast to previous vision-language tasks, UI modeling tasks are often concerned with a specific region or object on the screen, instead of the entire image. This unique aspect of UI modeling requires a model to be able to concentrate on a specific region on the image. We investigate different methods for realizing the Focus Region Extractor (see Figure 1) for computing region representation.

**Region-of-Interest (ROI) Align**: ROI Align is a parameter-free approach previously introduced for extracting a feature representation of a specific region on an image (He et al., 2017). This method performs average or max pooling over the feature map of an image region, and handles low spatial resolutions in the latent feature space using bilinear interpolation. In our case, we perform ROI Align for a target region over the image encodings from ViT outputs, which results in a fixed-length

vector representation for the focus region. This method provides a direct localization of the focus region in the encoded visual signals, and it embodies *soft* cropping as the resulted representation carries contextual information beyond the target area due to ViT self-attention. This is in contrast to *hard* cropping that carves out the raw pixels of the target area (Li et al., 2020b), which loses the access to the screen context.

**Region Summarizer**: Previous works have used learnable vectors as input in Transformer models to query visual modalities for specific tasks. For example, DeTR (Carion et al., 2020) used learnable queries for retrieving visual objects on an image. Flamingo (Alayrac et al., 2022) summarizes image encodings by using learnable query vectors to resample the image encodings. Inspired by these methods, we propose *Region Summarizer*, which acquires latent representation of a region based on ViT encodings, by using attention queries generated from the bounding box of the region or object. The bounding box of a region, $B \in \mathbb{R}^{4 \times 1}$, includes the four scalar coordinate values, i.e., [left, top, right, bottom]. For each coordinate value, we generate $n$ vectors (Equation 1), and having multiple descriptor vectors allows us to control the capacity of the region representation.

$$
\begin{aligned}
E &= \text{Reshape}_{4 \times nd_e \to 4 \times n \times d_e}([\text{GeLU}(BW_e)]W_x) \\
X &= \text{Reshape}_{4 \times n \times d \to 4n \times d}(E + C)
\end{aligned}
\tag{1}
$$

In Equation 1, we first project each coordinate—a scalar value—of the bounding box to a $nd_e$-dimensional dense vector using a perceptron: $\text{GeLU}(BW_e)$, where $W_e \in \mathbb{R}^{1 \times nd_e}$. We next reshape the perceptron activation from $4 \times nd_e$ to $4 \times n \times d_e$ to spawn $n$ $d_e$-dimensional vectors for each of the four coordinates. Then, $W_x \in \mathbb{R}^{d_e \times d}$ projects each vector to the hidden dimension, $d$, of the Focus Extractor transformer model. Thus $E \in \mathbb{R}^{4 \times n \times d}$ and each of the 4 coordinates produces $n$ $d$-dimensional vectors. To indicate from which coordinate value a vector comes, we add the coordinate type embedding, $C \in \mathbb{R}^{4 \times 1 \times d}$. Each vector in $C$ represents the embedding of a coordinate type in {left, top, right, bottom}. In sum, $W_e$, $W_x$ and $C$ are learnable parameters. $n$ is a hyperparameter that controls the number of bounding box vectors to use and the number of region summaries to generate, which determines the capacity of bottleneck.

Based on the bounding box descriptor vectors, $X$, we then use a Transformer to extract region representations from ViT encodings, $H \in \mathbb{R}^{m \times d}$, in a similar way to Flamingo (Alayrac et al., 2022). The query of cross attention $q = Q_i$ and the memory $kv = H \parallel Q_i \in \mathbb{R}^{(m+4n) \times d}$, which is the concatenation of $H$ and $Q_i$ on the first dimension. The $i$th layer of the Region Summarizer Transformer is formulated in Equation 2, where $Q_0 = X$ and $0 \le i \le l$ and $l$ is the number of Transformer layers used.

$$
\begin{aligned}
Y_{i+1} &= Q_i + \text{CrossAttention}(\text{q} = Q_i, \text{kv} = H \parallel Q_i) \\
Q_{i+1} &= Y_{i+1} + \text{Dense}(Y_{i+1})
\end{aligned}
\tag{2}
$$

where $Y_i \in \mathbb{R}^{n \times d}$ and $Q_i \in \mathbb{R}^{n \times d}$. As shown in our experimental analysis, the Region Summarizer allows Spotlight to focus on a target region indicated by a bounding box while taking into account of the screen context. Our experiments show that Region Summarizer offers superior performance over ROI Align. We conducted a series of ablations for Region Summarizer in Appendix B, and the pseudo code is shown in Appendix C.

We use an auto-regressive Transformer decoder to generate token sequences, by attending to the target region representation, $Q_l \in \mathbb{R}^{n \times d}$, via cross-attention. We formulate all the tasks as sequence decoding (see Figure 1). During pretraining, our model generates text content associated with a target region. For widget captioning and screen summary, they are naturally sequence decoding tasks. For command grounding and tappability prediction, given a target region, we ask the model to predict either Yes or No tokens following a prompt text, which can be a grounding command or a question for tappability. For command grounding, we let the model inspect each object on the screen by predicting Yes or No following a given command, and the object that yields the largest probability for the Yes token is used as the prediction. During both pretraining and finetuning, the entire Spotlight model is trained end to end by minimizing the cross entropy loss for next-token prediction during sequence decoding. Although there is no direct supervision for Region Summarizer, our analysis shows that the learned attention weights of Region Summarizer not only captures the target region but also relevant areas in the context of the screen (see Figure 2).

## 5 EXPERIMENTS

Similar to previous vision-language models (Chen et al., 2022; Alayrac et al., 2022), we use existing checkpoints of ViT and T5 models to initialize corresponding components in our models in all the experiments. We experiment with two different sized ViT models, B/16 and L/16, pretrained on 300M natural images (Dosovitskiy et al., 2021)[3]. ViT L/16 is larger than B/16 and has a similar parameter size as the mT5 base model (Xue et al., 2020), for which we also reuse the pretrained checkpoint[4]. The hyperparameters of the models and model variant details can be found in Appendix A.

### 5.1 PRETRAINING

During pretraining, we train the model to decode the text of an object in the screenshot. As this learning objective is generic, we can combine the heterogeneous C4 web dataset and mobile dataset by sampling objects and their text attributes. For C4 dataset, we randomly sample from three text attributes: `text` and `alt-text` as well as `misc-text` that includes all other text attributes, which occur much sparser than `text` and `alt-text`. We also sample the `title` attribute of a webpage with a small weight (0.01) to train the model with the ability to summarize the entire screenshot. For the mobile dataset, we randomly sample from the four text attributes of an object: `text`, `content_description`, `resource_id` and OCR text. We found that the text in these two datasets contains very frequent words or phrases, e.g., *submit* and *input password*. To counter the imbalance, we subsample the frequent phrases as in word2vec (Mikolov et al., 2013) with $t = 1e - 5$. As we want to explore few-shot prompting and multi-task learning in addition to task-specific finetuning, we sample a random number of $[1 - 5]$ screen-object-text tuples and pad the input sequence to 5 tuples, similar to Flamingo (Alayrac et al., 2022). As C4 dataset is much larger, we sample batches from C4 and mobile dataset with a ratio of 9:1. The ViT encoder encodes each screenshot independently, while the decoder decodes the full sequence of the text of all the pairs, concatenated by $BOC$ (Beginning of the chunk) and $EOC$ (End of the chunk) tokens. This is to encourage the model to generalize to using sequences of multiple screenshot-object-text tuples in multi-task and few-shot prompting scenarios for UI tasks, similar to previous work for the general domain.

### 5.2 FINETUNING, MULTI-TASK LEARNING AND FEW-SHOT PROMPTING

Once we pretrain the Spotlight model, we investigate three modeling strategies for downstream tasks, including task-specific finetuning, multi-task learning and few-shot prompting.

#### 5.2.1 FINETUNING

We finetune a pretrained Spotlight model for each of the four downstream tasks separately. Each example used for both finetuning training and evaluation contains a single screen-object-text tuple, because the model is tuned to perform the specific task and does not require task prompts or switchers. As shown in Table 3, we can see that our vision-only approaches obtain state-of-the-art performance for all the four tasks, significantly outperforming best previous models that use both screenshots and view hierarchies as input.

#### 5.2.2 MULTI-TASK LEARNING

Multi-task learning is valuable for reducing model footprint, but developing a robust multi-task model can be more difficult. We train a Spotlight model to learn the four tasks simultaneously, which is more challenging than finetuning for each specific task. We sample training batches from the widget captioning, screen summary, command grounding and tappability tasks with the weights of [3, 2, 15, 1], which are tuned on the development set. Each example in the batch contains two screen-object-text tuples during training so that the model is preconditioned to work with one prompt tuple and one prediction tuple. During evaluation, we applied the same model onto each of the four tasks with a screen-object-text tuple sampled from each training set to prompt the model to perform

---

[3]`https://github.com/google-research/vision_transformer`
[4]`https://github.com/google-research/multilingual-t5`

| | Model | Captioning | Summarization | Grounding | Tappability |
|---|---|---|---|---|---|
| Baselines | Widget Caption | 97.0 | - | - | - |
| | Screen2Words | - | 61.3 | - | - |
| | VUT | 99.3 | 65.6 | 82.1 | - |
| | Taperception | - | - | - | 85.5 |
| | Swearngin & Li (2019) | - | - | - | 87.9* |
| Spotlight | B/16 | 136.6 | 103.5 | 95.7 | 86.9 |
| | L/16 | **141.8** | **106.7** | **95.8** | **88.4** |

Table 3: Finetuning Performance. CIDEr scores are reported for widget captioning and screen summarization while accuracy for command grounding and F1 score for tappability prediction. * is obtained by Schoop et al. (2022) using both image and view hierarchy (Swearngin & Li, 2019).

different tasks. As shown in Table 4, multi-task Spotlight models still outperform previous models, and perform on par with task-specific models (Table 3) on widget captioning and tappability tasks. Although multi-task performance on screen summarization and grounding is lower than finetunning performance, they are still significantly better than the published benchmark results.

| Model | Captioning | Summarization | Grounding | Tappability |
|---|---|---|---|---|
| VUT multi-task | 99.3 | 65.1 | 80.8 | - |
| Spotlight B/16 | 140.0 | **102.7** | 90.8 | 89.4 |
| Spotlight L/16 | **141.3** | 99.2 | **94.2** | **89.5** |

Table 4: The performance of Spotlight models for multi-task learning.

### 5.2.3 FEW-SHOT LEARNING

Few-shot learning is an important capability demonstrated by large vision and language models (Alayrac et al., 2022; Brown et al., 2020), in which a large model can be adapted for a specific downstream task solely using examples or prompts without further training. For few-shot prompting, we use a pretrained Spotlight model directly for evaluation, without model parameter finetuning. We test Spotlight on widget captioning tasks with a different number of shots: $\{0, 4, 8, 16, 32\}$ to see how the model performance is impacted by the number of example prompts (Table 5). When there is 0 shot, the pretrained Spotlight model is directly tested on the widget captioning task, without example prompts. Although at most five screen-object-text tuples are used in each example during pretraining, our model can generalize to more shots during inference. We observed marginal improvement over the zero-shot condition when more shots of the task are given for the L/16 model, but not for the B/16 model, which implies that models with a larger capacity seems more likely to succeed on few shot learning. The models did not show meaningful few-shot performance for other tasks. We conjecture that the other tasks are too far from the pretraining data distribution, as our pretraining datasets are still much smaller than the ones in general domain (Alayrac et al., 2022). Using a larger and more diverse pretraining dataset and a larger text decoder might improve the few-shot learning performance for UI tasks.

| ViT | 0 | 4 | 8 | 16 | 32 |
|---|---|---|---|---|---|
| B/16 | 57.1 | 56.7 | 55.6 | 55.5 | 54.9 |
| L/16 | 61.6 | 61.9 | 62.0 | 61.9 | 62.1 |

Table 5: The performance of few-shot prompting on the Widget Captioning task.

## 6 DISCUSSIONS

To understand how the Region Summarizer is able to enable Spotlight to focus on a target region as well as relevant areas on the screen, we analyze the attention weights for both Widget Captioning

and Screen Summarization task. In the Widget Captioning example (Figure 2a), we can see the model learns to attend to not only the target region of the check box, but also the text "Chelsea" on the far left to generate the caption. For the Screen Summarization example in Figure 2b where the target region is the entire screen, the model learns to attend to important parts on the screen for summarization.

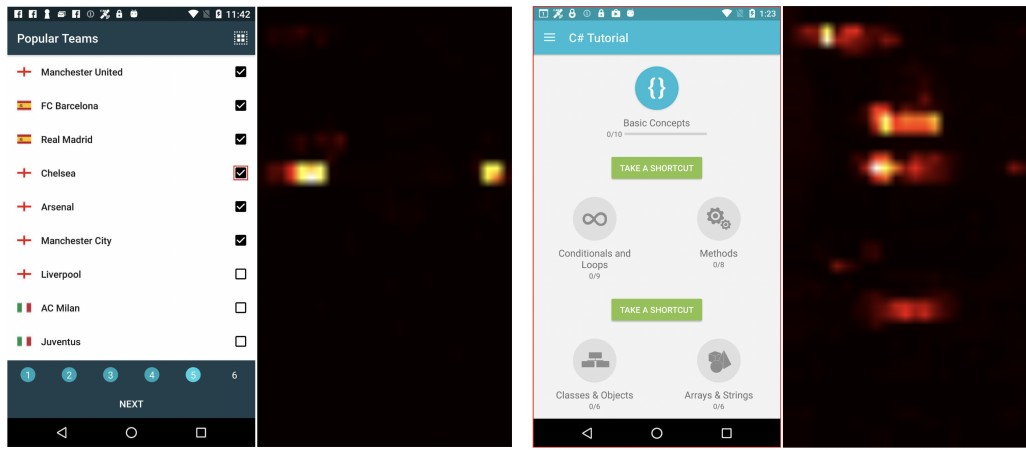

(a) Predicted caption: *select chelsea team*

(b) Predicted summary: *page displaying the tutorial of a learning app*

Figure 2: Visualization of the attention weights of the last cross-attention layer of the Region Summarizer in the L/16 fintuned models for Widget Captioning (a) and Screen Summarization (b).

Spotlight models advance the art of UI understanding by using a large-scale pretraining dataset and a general architecture that can support different use cases. As shown in our ablation study (Appendix B), pretrained models significantly outperformed the models trained from scratch. There are various ways of using the unlabeled dataset to pretrain UI models (He et al., 2020; Bai et al., 2021). In our early exploration, we also experimented with decoding a serialized view hierarchy directly instead of only text of individual objects. However, this led to poor pretraining performance, likely due to the large amount of noise in view hierarchy structures (Li et al., 2022). Nevertheless, the structural information encapsulated in the view hierarchy is valuable and investigating the usage of such information for better pretraining remains an interesting direction for future work.

Compared to recent large vision-language model efforts (Alayrac et al., 2022; Chen et al., 2022) that use $O(10B)$ parameters, the model size that we investigated is relatively small (Appendix A). Our experiments consistently show the trend of larger model yielding better performance, which promises that simply scaling up the model sizes would potentially lead to further performance gains. For this purpose, Spotlight can easily adopt larger ViT and T5 checkpoints. Our model architecture is generic, which takes a screenshot and a region bounding box of interest as well as text prompts, and outputs text responses. Spotlight can be easily applied to more UI tasks, e.g., icon classification (Deka et al., 2017) and multi-step grounding (Li et al., 2020a; Burns et al., 2022), as many UI tasks can be formulated as a sequence decoding task.

## 7 CONCLUSIONS

We presented Spotlight, a vision-only approach for mobile UI understanding, which alleviates the need to use view hierarchy. We design Spotlight based on a general vision-language model architecture and enable the model to focus on a region of interest on the screen. This architecture is easy to scale and can benefit from the success of recent large vision-language models pretrained for the general domain. Spotlight is able to fulfill a range of UI modeling tasks. Our experiments show that Spotlight obtains SoTA results on several representative UI tasks and outperforms previous works that use both screens and view hierarchies. We conduct ablations on a range of model variants, and report our findings on these variants, as well as a variety of modeling strategies and use cases, including finetuning, multi-tasking learning and few-shot prompting.

ACKNOWLEDGEMENT

We thank the anonymous reviewers for their constructive feedback for improving the paper. We thank Mandar Joshi and Tao Li for their help in processing the C4 dataset. We also thank Chin-Yi Cheng and Forrest Huang for their feedback.

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

## A Hyperparameters & Model Sizes

We base Spotlight models in our experiments on T5 and ViT. We used two ViT configurations: B/16 and L/16 (Table 6). We use an image resolution $[740, 740]$ for all our experiments. The mobile screenshots are resized with aspect ratio preserved and padded to $[740, 740]$ if necessary. The paddings in the pixels are ignored by the ViT encoder using masking. The object bounding boxes are normalized to $[0, 1]$. We use 0.1 dropout in all our models. For all the training experiments, we use a batch size of 128 and linear warmup/rsqrt decay for learning rate. During inference decoding, a beam size of 5 is used. The maximum text decoding length is 64 for each screen-object-text tuple.

| T5 | ViT | ViT Enc Layers | Dec Layers | # Parameters |
|------|------|----------------|------------|--------------|
| base | B/16 | 12 | 12 | 619M |
| base | L/16 | 24 | 12 | 843M |

Table 6: The Spotlight model variants in our experiments.

**Region Summarizer**: For both the B/16 and L/16 model, we use $n = 4, d_e = 64, d = 768$ for Equation 1 and 2.

**Pretraining**: We pretrain the Spotlight models with a learning rate of 9e-3 for both B/16 (164K steps) and L/16 models (156K steps), with an initial linear warmup to 10k steps. Each training example consists of $[1, 5]$ image-object-text tuples sampled from the pretraining datasets. The model with ViT B/16 is trained using 128 Google Cloud TPU v3 cores for 54 hours. The model with ViT L/16 is trained using 256 Google Cloud TPU v3 cores for 86 hours.

**Finetuning**: For finetuning, we use a learning rate of 1e-3 and 20k steps for the Command Grounding task and 1e-4 and 10k steps for the other three tasks.

**Multi-Task Learning**: We use a learning rate of 3e-4 for multi-task learning, and train the multi-task Spotlight models for 30k steps. The sampling weights are $[3, 2, 15, 1]$ for the Widget Captioning, Screen Summarization, Command Grounding and Tappability tasks. We use 2 screen-object-text tuples in each example during training. During evaluation, a screen-object-text tuple from the training set of the target task is used to prompt the model to perform the task.

## B  ABLATION STUDY

To explore different options for designing the Spotlight models, we conduct an extensive ablation study. We use the model with B/16 ViT as the full model. Each ablated version has one option changed compared to the full model, which is listed in the first column in Table 7. We pretrain all the ablation variants for 100K steps and finetune them on the four downstream tasks using the same hyperparameters and training schedule (see Appendix A). The full model in Table 3 has better results as it is pretrained with more steps (164K).

Most notably, the models that use a frozen ViT or are trained from scratch did not perform well for all the four tasks. This indicates the pretraining on the mobile and web datasets is crucial for the downstream UI tasks. Using only one of the two pretraining datasets also led to degraded performance, which demonstrates the value of combining heterogeneous datasets.

Region Summarizer is an important component that we proposed for enabling Spotlight to focus on a specific region on the screen. We compare a few options for the design of Region Summarizer. In Equation 2, the region representation is concatenated with ViT encodings to form the memory for cross attention and the region representation is updated after each layer: $kv = H \parallel Q_i$ and $Q_0 = X$. We investigate a static region representation in kv, i.e., $kv = H \parallel X$, which is not updated by each cross-attention layer and no region representation kv, i.e., $kv = H$. We also investigate embedding four bounding box coordinates jointly instead of individually. These alternatives perform reasonably on the UI tasks except the Widget Captioning task. This indicates that more flexibility in the bounding box queries, e.g., embedding each coordinate individually and updating its embedding using cross-attention per layer, helps the Widget Captioning task. In the Section 6, we can see that the bounding box queries not only attend to the area of the bounding box itself, but also the relevant context on the screen needed for decoding the captions.

Lastly, using ROI Align directly performed worse for the screen summarization task than using Region Summarizer, possibly due to that some information of the entire screen can be lost during the average pooling. In contrast, using the vector from ROI Align as the query to Region Summarizer instead of the bounding box led to improved results for text decoding tasks, but performed worse for the Command Grounding task.

| Ablation | Widget Caption | Screen Summary | Grounding | Tappability |
|---|---|---|---|---|
| Full model | 125.1 | 95.7 | 95.6 | 86.9 |
| Freeze ViT | 37.5 | 19.6 | 72.6 | 72.3 |
| From scratch | 18.5 | 23.1 | 36.7 | 79.5 |
| C4 only | 120.3 | 94.2 | 96.3 | 87.8 |
| Mobile only | 105.7 | 90.3 | 89.8 | 87.3 |
| Static bbox queries | 118.0 | 96.2 | 95.1 | 87.4 |
| No bbox in KV | 114.7 | 93.7 | 95.9 | 87.8 |
| Joint bbox embedding | 119.0 | 94.0 | 96.7 | 87.8 |
| ROI Align as query | 124.4 | 94.9 | 89.4 | 87.4 |
| ROI Align | 121.8 | 87.5 | 89.0 | 86.8 |

Table 7: Ablation study results.

## C  PSEUDO CODE

```
def region_summarizer(
    bbox,          # The bounding box coordinates in [batch, 4, 1].
    vit_outputs,   # The ViT outputs in [batch, vit_seq_len, hidden_dim].
    num_layers,    # Number of cross-attention layers.
    hidden_dim,    # Hidden dimension for the cross-attention layers.
    output_dim,    # Output dimension for the first dense layer.
    num_query,     # The number of queries generated from each coordinate.
):
  # First, map bounding box coordinates to dense vectors.
  bbox = gelu(Dense(bbox, features=output_dim))
  # Split each bbox vector into multiple queries.
  bbox = bbox.reshape([batch, 4, num_query, -1]
  # Map to the dimension required by cross-attention layers.
  bbox = Dense(bbox, features=hidden_dim)

  # Compute type embeddings for the four coordinates.
  coord_types = [[1], [2], [3], [4]]
  type_embedding = Embed(coord_types)
  # Reshape for proper broadcast.
  type_embedding = type_embedding.reshape([1, 4, 1, hidden_dim]

  # Combine coordinates with the type embeddings.
  bbox = bbox + type_embedding

  # Perform cross-attention to extract region information.
  x = bbox
  for i in range(num_layers):
    kv = concat([vit_outputs, x], axis=1)
    x = x + CrossAttention_i(q=x, kv)
    x = x + Dense_i(x)
  return x
```

Listing 1: Region Summarizer Psuedo Code

## D  PRETRAINING DATASETS

In the mobile pretraining dataset, view hierarchy is a tree structure of a mobile UI, like the DOM tree to a webpage. In a view hierarchy, each leaf node corresponds to an object on the screen, which contains a collection of attributes such as object type, text content, visibility and clickability. Each non-terminal node represents a group of objects. However, in reality, view hierarchies are often acquired from the runtime render tree of a UI screen, which can contain much noise and might not be aligned with the visual representation.

| Attribute | Count | Avg word # |
|---|---|---|
| title | 2.6B | 10.7 |
| text | 2.3B | 5.1 |
| alt-text | 136.1M | 3.8 |
| aria-label | 90.4M | 2.7 |
| placeholder | 33.1M | 2.2 |
| data-icon | 7.2M | 1.5 |
| data-service | 874K | 1.1 |
| data-caption | 241K | 9.7 |
| uk-icon | 167K | 1.6 |
| data-hint | 134K | 3.6 |
| data-svg | 126K | 1.6 |
| data-social | 114K | 1.9 |
| uk-tooltip | 68K | 4.0 |
| voq-icon | 4K | 3.0 |

Table 8: Statistics of the HTML attributes in the C4 pretraining dataset.

We list all the HTML attributes used for pretraining and their statistics in the C4 dataset (Table 8). For both the web and mobile dataset, we remove invisible objects (based on CSS or view hierarchy attributes) and objects with bounding box of uniform color, and filter out object text consisting of only generic words. The list of generic words are the following: action, bar, menu, title, and, ans, app, icon, name, arg, background, element, btn, but, bottom, button, content, desc, text, item, empty, fab, image, grid, header, img, imgfile, lbutton, label, letter, list, view, pic, placeholder, random, row, single, raw, small, large, sub, template, navbar, banner, test, textinput, error, texto, todo, toolbar, tool, track, txt, unknown, stub, web, left, right, tlb, nan, page, feature, menugrid, picture, tabs, number, node, iconimage, entity, webview, heading, logo, tbl, tab, primary, footer. We also remove object text that contains only single characters or URLs. Continuous spaces and underscores are replaced with a single space. All the text are lowercased.

## E  DOWNSTREAM TASKS & DATASETS

We list the dataset statistics of the four downstream UI tasks in Table 9 and show one example of each task in Figure 3. For Widget Captioning, Command Grounding and Tappability, the models use individual UI objects, e.g., decoding the caption of an UI object. For Screen Summarization, the models are trained to decode the summary of an entire screen. More detailed information of the four datasets can be found in their original papers.

| Task | Split | Screens | UI Objects |
|---|---|---|---|
| Widget Captioning  (Li et al., 2020b) | Train | 14,878 | 41,221 |
| | Dev | 1,292 | 3,483 |
| | Test | 1,265 | 3,621 |
| Screen Summarization (Wang et al., 2021) | Train | 15,743 | - |
| | Dev | 2,364 | - |
| | Test | 4,310 | - |
| Command Grounding (Li et al., 2021) | Train | 7,822 | 178,783 |
| | Dev | 1,024 | 24,007 |
| | Test | 987 | 22,974 |
| Tappability (Schoop et al., 2022) | Train | 2,567 | 14,781 |
| | Dev | 309 | 1,857 |
| | Test | 342 | 2,029 |

Table 9: The dataset statistics of the four downstream tasks.

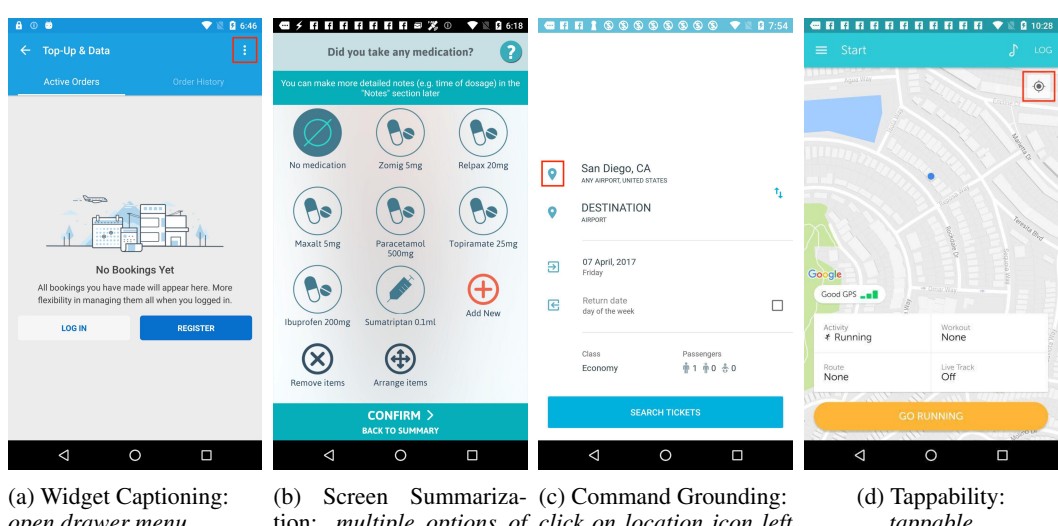

(a) Widget Captioning: *open drawer menu*

(b) Screen Summarization: *multiple options of medicine in health app*

(c) Command Grounding: *click on location icon left to san diego ca*

(d) Tappability: *tappable*

Figure 3: Examples of the four UI tasks. For Widget Captioning, Command Grounding and Tappability Prediction, the target object is highlighted for illustration purposes only, i.e., these red bounding boxes are not marked on the images that are fed to the model.

