# OpenReview forum: "Spotlight: Mobile UI Understanding using Vision-Language Models with a Focus"
_ICLR.cc/2023/Conference — ICLR 2023 poster_

### Official Review · Reviewer_WFS1 · 2022-10-24

**Confidence:** 3
**Correctness:** 4
**Technical Novelty And Significance:** 3
**Empirical Novelty And Significance:** 3
**Recommendation:** 5

**Clarity, Quality, Novelty And Reproducibility:**

Clarity: As discussed in the Strength and Weakness, I would highly recommend heavy re-writing to ensure the best possible impact of the great technical results. Currently it is hard to follow the story and understand some of the technical details.

Quality: The proposed model demonstrated substantial accuracy improvement over previous models. The ablation is solid in showing key contributing factos such as region summarizer and pretraining. I consider the experiment quality as high.

Novelty: Given the authors challenged a common believe on view-hierarchy info based model is a hands-down choice for good accuracy, and demonstrate SOTA accuracy using visual input only model. I would rate the novelty as high.

**Strength And Weaknesses:**

Strength

1. The technical effort is solid. The creation of the pretrained model involves the collection and curation of millions of UI screen shots. Such endeavor combined with the novel pretraining objective for UI understanding task enables doubling the CIDEr score from no pretraining efforts on the same model.

2. The empirical results is impressive. It attains SoTA accuracy on the 3 tasks with only visual info input during inference time.

Weakness

The paper is a bit hard to follow and needs modification to enable a self-containing manuscript for readers to understand.

1. There are technical concept/facts missing their intuitive or formal definitions in the related work / background sections. Representative missing information are
     a. What is an intuitive definition of view-hierarchy information which is the primary input to existing models.
     b. How severe is the information missing problem in view-hierarchy information? Could you slightly quantify it w.r.t. the datasets the author used or collected? This is important to evaluate the claim on the problem of previous view-hierarchy info based models.

2. The model design is a bit hard to fully understanding with pure text. Overall, I would recommend a symbol based formal model description which could answer the following questions.
    a. In listing 1, it seems the Dense function encodes the coordinate into vectors. Is this encoding the normal way in VIT which just maps the coordinates to the 1D index of the image patch in the patch sequence?
    b. Also why the coordinate is encoded twice via both the Dense function and the SinusoidalEmbed function?
    c. What is the training objective? My rough guess from the relevant description in 4.1 is that it use the self contrastive loss type thing to promote the patch and text description pairs. But it is very unclear to me if this is the case.

**Summary Of The Paper:**

The authors introduced a novel vision-language multi modality model (Spotlight) for mobile UI understanding tasks. Specifically this model is designed for tasks for mapping visual components in the UI to natural language text. In the context of UI understanding models, this paper presents 2 major contributions.

1. The authors challenge the recent belief that view-hierarchy based UI understanding model is a hands-down choice for accuracy. They proposed the Spotlight model which only uses visual information at inference time; this can potentially lead to higher accuracy by working around the typical incomplete hierarchical view information for prediction.

2. The authors developed a pretrained model for the UI understanding tasks. They present SoTA accuracy on UI understanding tasks under single-task finetuning, single-task few shot and multi-task finetuning. They show that the pretraining is crucial to the high accuracy in widget captioning, screen summarization, command grounding tasks.

**Summary Of The Review:**

Overall I vote for a marginally-below-acceptance. I think the empirical results are overall great and the visual input only model is a novel design over the conventional view-hierarchy based models [Based on my limited understanding of UI view-hierarchy]. However I think the writing quality is below the bar for top tier conference submissions. Disclaimer: :) I am experienced in developing large language model and multi modality models. But I am not an expert in the UI understanding domain.

---

> ### Author Response · Authors · 2022-11-18
> **Responses to Reviewer WFS1**
>
> Thank you very much for your detailed reviews. We apologize for the omission of several details. We have addressed all your concerns in the revision, and improved the paper by including further technical details, new analysis, additional ablations and an additional UI task. We here respond to each of your questions, and please let us know if you have any further questions.
>
> > 1. There are technical concept/facts missing their intuitive or formal definitions in the related work / background sections. Representative missing information are a. What is an intuitive definition of view-hierarchy information which is the primary input to existing models. b. How severe is the information missing problem in view-hierarchy information? Could you slightly quantify it w.r.t. the datasets the author used or collected? This is important to evaluate the claim on the problem of previous view-hierarchy info based models.
>
> ### Re: a
>
> View hierarchy is a tree structure of a mobile UI, like the DOM tree to a webpage. In view hierarchy, each leaf node corresponds to an object on the screen, which contains a collection of attributes such as object type, text content, visibility and clickability. Each non-terminal node represents a group of objects. However, in reality, view hierarchies are often acquired from the runtime render tree of a UI screen, which can contain much noise and might not be aligned with the visual representation.
>
> ### Re: b
>
> Previous studies (Ross et al 2018, Li et al 2022) revealed that a large portion of view hierarchy has missing text information or misaligned structure-visual representation. In particular, about 37.4% of the screen view hierarchies contain objects with invalid bounding boxes. 92.0% of Floating Action Buttons, 54.7% of Image Buttons and 86.3% of Clickable Images had missing text labels, while text features are important for the multi-modal models. Removing text features resulted in a drop of 17 CiDER points for the Widget Captioning model (Li et al 2020). We have added this information to the Introduction for better motivation.
>
> > 2. The model design is a bit hard to fully understanding with pure text. Overall, I would recommend a symbol based formal model description which could answer the following questions. a. In listing 1, it seems the Dense function encodes the coordinate into vectors. Is this encoding the normal way in VIT which just maps the coordinates to the 1D index of the image patch in the patch sequence? b. Also why the coordinate is encoded twice via both the Dense function and the SinusoidalEmbed function? c. What is the training objective? My rough guess from the relevant description in 4.1 is that it use the self contrastive loss type thing to promote the patch and text description pairs. But it is very unclear to me if this is the case.
>
> ### Re: presentation clarity
>
> Sorry again for the issue with the presentation in our early version. In the revision, we have substantially improved the description of Region Summarizer, including detailed math equations in Section 4 (as you suggested), more complete Pseudo code in the Appendix C, and extensive ablation of the design of Region Summarizer in Appendix B. Furthermore, we added an analysis on attention weights of bounding box queries in Section 6, which reveals that Region Summarizer not only attends to the region indicated by the bounding box but also the relevant areas on the screen context for achieving a downstream task. Please let us know if there are any unclear details.
>
> ### Re: training objective
>
> The training objective is minimizing next token cross-entropy loss as in the language models. We have clarified this at the end of Section 4.

---

> > ### Author Response · Authors · 2022-12-05
> > **Please examine our revision and responses. Thanks!**
> >
> > Dear Reviewer,
> >
> > We have substantially improved the writing quality that you pointed out. We really hope you can inspect our revision and responses, which have addressed the issues you pointed out. Thanks!
> >
> > Authors of Paper5558

---

### Official Review · Reviewer_cKCz · 2022-10-24

**Confidence:** 4
**Clarity, Quality, Novelty And Reproducibility:** It is a original work but some detail…
**Correctness:** 4
**Technical Novelty And Significance:** 3
**Empirical Novelty And Significance:** 3
**Recommendation:** 6

**Strength And Weaknesses:**

Strength:
The authors attempted to study UI understanding tasks by multi-task learning with pretrained multi-modal models, which is an independent, innovative and laborious work. The experiment results prove the generalization and validity of this method. Meanwhile, The author also proposes a large-scale UI modeling pretraining dataset.

Weakness:
1. Although the motivation is innovative in constructing the pretraining model for the UI modeling field, the overall pretraining pipeline may lack appropriate innovation and some aspects are similar to Flamingo, e.g., the vision-language model architecture, the evaluation method on the multi-task and few-shot learning, and the computational pipeline of Region Summarizer is similar to the Perceiver Resampler of Flamingo.
2. The illustration of the pretraining dataset is too brief. It is recommended to add more statistical analysis and construction details, e.g., The text and content description are human-annotated or not?
3. Considering the readability, it is recommended to add the introduction of test datasets about downstream tasks. Perhaps the authors can add these in the supplementary material rather than suggesting the readers read the previous literature.
4. The ablation study is limited and maybe authors can consider the following ablation to make the paper clear and complete.
(1)	Region Summarizer:
①	Why choose the bbox coordinates as q rather than the region feature of bbox?
②	What is the result when kv is just the vit_outputs rather than the concatenation of vit-outputs and bbox.
(2)	Pretraining:
①	Can the text input is concatenated by the four text elements of an object?
②	The ViT is freezing as Flamingo or not?  If not, The much screenshots of examples will bring the burden of GPU memory or not?
5. The 2.69M pre-training data as extra knowledge may obscure the fairness of the method compared with the baseline method. Maybe author can try to further finetune the comparison methods combined with single-modal encoders of the pretraining model? This is just a suggestion out of curiosity and not influence the final judgment.
6. In the Discussion section, these “early exploration” are supposed to illustrate hypotheses by some experimental data.
7. The paper of the comparison method “Widget caption” is cited repeatedly.
8. Maybe the authors can consider making the dataset and code public to promote the development of this field. It is just advice and not influence the final judgment.


**Summary Of The Paper:**

For the mobile UI understanding tasks, the authors achieve advanced performance on multi-task learning and few-shot learning by pretraining the vision-language model on the proposed 2.69M dataset. This pretraining-finetune framework is easily scalable to other UI modeling tasks and not needs the view hierarchy as auxiliary information. Meanwhile, the authors proposed a pipeline for automatically creating the large-scale UI understanding dataset.

**Summary Of The Review:**

It is recommended to add some details and ablation study in paper. Moreover, since there is very little research in this field, I encourage the authors to open source the pretraining dataset,model checkpoints, and code, which will promote the development of this field and attract more researchers to make the field more prosperous.

---

> ### Author Response · Authors · 2022-11-18
> **Responses to Reviewer cKCz**
>
> Thank you very much for your insightful comments and encouraging feedback. We have improved the paper substantially to address your concerns and beyond, including further technical details, new analysis, additional ablations and an additional UI task, which have been reflected in the revision. We here respond to each of your questions, and please let us know if you have any further questions.
>
> > 1. Although the motivation is innovative in constructing the pretraining model for the UI modeling field, the overall pretraining pipeline may lack appropriate innovation and some aspects are similar to Flamingo, …, and the computational pipeline of Region Summarizer is similar to the Perceiver Resampler of Flamingo.
>
> Although Spotlight is built on top of Flamingo, our work investigates a new domain and produces new findings. In particular, we proposed new methods for region representations, which is critical for UI modeling tasks. In the revision, we have added further details of Region Summarizer, including detailed math equations in Section 4, more complete Pseudo code in the Appendix C, and extensive ablation of the design of Region Summarizer in Appendix B. Furthermore, we added an analysis on attention weights of bounding box queries in Section 6, which reveals that Region Summarizer not only attends to the region indicated by the bounding box but also the relevant areas on the screen context for achieving a downstream task.
>
> > 2. The illustration of the pretraining dataset is too brief. It is recommended to add more statistical analysis and construction details, e.g., The text and content description are human-annotated or not?
>
> We have added more statistics of pretraining datasets in Section 3.1, including a newly added C4 dataset that is publicly available and improves the performance, and additional details in Appendix D. For pretraining datasets, the text and content description are retrieved from the web pages or view hierarchies, which are automatically collected and not annotated by human.
>
> > 3. Considering the readability, it is recommended to add the introduction of test datasets about downstream tasks…rather than suggesting the readers read the previous literature.
>
> We have added more detailed statistics and examples for the downstream tasks in Appendix E.
>
> > 4. The ablation study is limited and maybe authors can consider the following ablation to make the paper clear and complete. (1) Region Summarizer: ① Why choose the bbox coordinates as q rather than the region feature of bbox? ② What is the result when kv is just the vit_outputs rather than the concatenation of vit-outputs and bbox. (2) Pretraining: ① Can the text input is concatenated by the four text elements of an object? ② The ViT is freezing as Flamingo or not? If not, The much screenshots of examples will bring the burden of GPU memory or not?
>
> We have conducted additional ablation as you suggested (Appendix B). The ablation study shows 1) ROI feature instead of bounding box as query led to weaker results for grounding, 2) The model without bbox in KV performed worse on widget captioning and screen summarization, 3) Frozen ViT led to poor performance, although the training can be faster as you pointed out. We did not concatenate the four text elements as each of them often describes different aspects of the element. Concatenation also enforces a sequential order between them, which might not be valid.
>
> | Ablation | Widget Caption | Screen Summary | Grounding | Tappability |
> | - | - | - | - |- |
> |    Full model | 125.1 | 95.7 | 95.6 | 86.9 |
> |    ROI Align as query | 124.4 | 94.9 | 89.4 | 87.4 |
> |    No bbox in KV | 114.7 | 93.7 | 95.9 | 87.8 |
> |    Freeze ViT | 37.5 | 19.6 | 72.6 | 72.3 |
>
> > 5. The 2.69M pre-training data as extra knowledge … This is just a suggestion out of curiosity and not influence the final judgment.
>
> We agree the investigation you suggested will further complete our understanding. In this work, our goal is to investigate the feasibility of removing the dependency on view hierarchy and the potential for vision-only approaches.
>
> > 6. In the Discussion section, these “early exploration” are supposed to illustrate hypotheses by some experimental data.
>
> We experimented with pretraining by decoding the entire view hierarchy, inspired by Pix2seq [Chen et al. ICLR’22]. However, the approach performed poorly on UI tasks, and did not yield meaningful results. We think the model is overwhelmed by the noise in view hierarchy, which does not always align with visual representation.
>
> > 7. The paper of the comparison method “Widget caption” is cited repeatedly.
>
> We have cleaned the citations.
>
> > 8. Maybe the authors can consider making the dataset and code public to promote the development of this field. It is just advice and not influence the final judgment.
>
> We will open source the mode and data pipeline code. The C4 dataset that we used for pretraining is already public and we will pursue the opensource of the mobile data.

---

> > ### Author Response · Authors · 2022-12-05
> > **Please examine our revision and responses. Thanks!**
> >
> > Dear Reviewer,
> >
> > Thank you again for your insightful reviews. We have significantly improved the paper based on your comments. Please let us know if you have any further questions.
> >
> > Authors of Paper5558

---

### Official Review · Reviewer_V2L4 · 2022-10-24

**Confidence:** 4
**Clarity, Quality, Novelty And Reproducibility:** Please see the comments above.
**Correctness:** 3
**Technical Novelty And Significance:** 2
**Empirical Novelty And Significance:** 2
**Recommendation:** 5

**Strength And Weaknesses:**

Strength
1. The idea of incorporating bounding box queries to obtain region-specific features provides a unique way to extract fine-grained features.
2. Compared with existing approaches, impressive performance on finetuning experiments for three mobile UI understanding tasks were achieved.

Weaknesses:
1. Overall, the novelty of this work is limited. The model architecture is similar to Flamingo (Alayrac et al., 2022). The only difference is that the inputs of Flamingo are visual tokens and learned latent queries, while SPOTLIGHT takes visual tokens and bounding box queries.
2. The main idea of this work is that, the bounding box queries have capacities to learn how to represent specific image regions, while jointly taking into account of the screen context. However, there is no further analysis, explanation, or qualitative/quantitative supports for this claim.
3. The experiment comparisons are potentially unfair. The improved results reported by SPOTLIGHT might come from large-scale pretraining, overwhelming parameters, and the uses of more powerful models (i.e., ViT and T5). All the baseline models (e.g. Widget Caption (Li et al., 2020b) and Screen2Words (Wang et al., 2021)) are composed of ResNet and vanilla Transformer with 128 dimensions.
4. Missing comparisons to SOTAs in multi-task (Table 4) and few-shot learning tasks (Table 5).
5. In Section1, the paper states its second contribution: “we develop a method for creating a large-scale pretraining dataset from automatically collected mobile screens.” However, this newly dataset is not discussed or presented later.
6. In Sect. 5.3, the authors note that SPOTLIGHT outperforms previous models. However, in Table 4, there are no other experiments to support this claim.








**Summary Of The Paper:**

This paper proposes a model that incorporates bounding box queries in the visual encoder to extract region-specific features for UI modeling tasks. With the proposed framework, no view hierarchy would be required during the above modeling process, which thus alleviates the potential concerns of missing object text or bounding box misalignment.


**Summary Of The Review:**

This paper proposes a model that incorporates bounding box queries in the visual encoder to extract region-specific features for UI modeling tasks. With the proposed framework, no view hierarchy would be required during the above modeling process, which thus alleviates the potential concerns of missing object text or bounding box misalignment. However, the paper is not well presented (with insufficient technical contributions and details, and missing or over-claimed experimental results). I would reconsider my ratings if the above issues can be properly addressed.

---

> ### Author Response · Authors · 2022-11-18
> **Responses to Reviewer V2L4**
>
> Thank you very much for your detailed comments and constructive feedback for improving the paper. To address your concerns, we have revised the paper thoroughly, including further technical details, new analysis, additional ablations and an additional UI task. We respond to each of your questions here and have made these improvements in the revision.
>
> > 1. Overall, the novelty of this work is limited. The model architecture is similar to Flamingo (Alayrac et al., 2022). The only difference is that the inputs of Flamingo are visual tokens and learned latent queries, while SPOTLIGHT takes visual tokens and bounding box queries.
> 2. The main idea of this work is that, the bounding box queries have capacities to learn how to represent specific image regions, while jointly taking into account of the screen context. However, there is no further analysis, explanation, or qualitative/quantitative supports for this claim.
>
> We have significantly improved the presentation of the work, and addressed these concerns with the following changes.
> * We strengthened our novelty by adding more details of our Region Summarizer, including detailed math equations in Section 4 and more complete Pseudo code in the Appendix C.
> * To understand how Region Summarizer enables the model to focus on the target region and take into account the screen context, we added an analysis on attention weights of bounding box queries over ViT encodings in Section 6. Our analysis reveals interesting attention patterns, which show that Region Summarizer not only attends to the region indicated by the bounding box query but also the relevant areas on the screen (the context) for achieving a downstream task.
> * We conducted extensive ablation study (Appendix B) to show the effectiveness of the proposed method compared to other choices of region representations, e.g., ROI align versus Region Summarizer, ROI align as query versus bounding box as query in Region Summarizer, bounding box coordinates embedded separately versus jointly, and bounding box queries remaining the same versus updated each layer.
>
> > 3. The experiment comparisons are potentially unfair. The improved results reported by SPOTLIGHT might come from large-scale pretraining, overwhelming parameters, and the uses of more powerful models (i.e., ViT and T5). All the baseline models (e.g. Widget Caption (Li et al., 2020b) and Screen2Words (Wang et al., 2021)) are composed of ResNet and vanilla Transformer with 128 dimensions.
>
> We agree our models are much larger, which is a key factor for vision-language models to obtain good performance, e.g., Flamingo (Alayrac et al 2022) and PALI (Chen et al 2022). The baseline models rely on view hierarchy, which can help the small models to learn quickly. However, these baseline models are difficult to scale due to their task-specific architectures and data constraints (e.g., requiring view hierarchy), which makes it difficult for these baseline models to achieve high accuracy and generalization across tasks. Spotlight investigates vision-only approaches for UI modeling and demonstrates the potential that large models can achieve. We feel this is a valuable exploration and contributes to the field with new knowledge about how large vision-language models can perform for the fast developing domain of UI modeling.
>
> > 4. Missing comparisons to SOTAs in multi-task (Table 4) and few-shot learning tasks (Table 5).
>
> Thanks for pointing it out. We have added previous SOTA for multi-task learning for easy comparison. For few-shot prompting, we are the first to try this setting for UI tasks and there is no previous baseline.
>
> > 5. In Section1, the paper states its second contribution: “we develop a method for creating a large-scale pretraining dataset from automatically collected mobile screens.” However, this newly dataset is not discussed or presented later.
>
> We have strengthened Section 3.1 to include more details and statistics of the mobile pretraining dataset, and added further details in Appendix D. In our latest experiments, we have increased image resolutions and added new pretraining data based on the publicly available C4 dataset, which lead to even better performance for all the UI tasks. We feel our method along with these details for acquiring large datasets can benefit others for scaling up their datasets and pursuing large-scale UI modeling in the future.
>
> > 6. In Sect. 5.3, the authors note that SPOTLIGHT outperforms previous models. However, in Table 4, there are no other experiments to support this claim.
>
> We have added previous SOTA for multi-task learning for easy comparison in Table 4.
>
> > However, the paper is not well presented (with insufficient technical contributions and details, and missing or over-claimed experimental results).
>
> We have comprehensively improved the paper by addressing your concerns and adding additional experiments, analysis and technical details. Please let us know if you have any further questions for us to address.

---

> > ### Author Response · Authors · 2022-12-05
> > **Please examine our revision and responses. Thanks!**
> >
> > Dear Reviewer,
> >
> > Thank you again for your thorough reviews and constructive feedback. We really hope you can take a look at our revision and responses, which we believe have addressed the issues you raised.
> >
> > Authors of Paper5558

---

> > > ### Comment · Reviewer_V2L4 · 2022-12-11
> > > **Follow-ups**
> > >
> > > Many thanks for providing the response to my previous concerns. This paper is potentially interesting. However, with concerns and possible improvements on technical contributions and completeness of experiments, I feel that this paper is immature to be published now. Therefore, I will keep my rating.

---

> > > > ### Author Response · Authors · 2022-12-12
> > > > **Thanks for your further feedback**
> > > >
> > > > Thank you for your follow-up feedback. Our updated paper revision has many improvements. Particularly, we strengthened our technical contribution with more details and better presentation. Our experiments are also more complete now with many additional ablations and analysis. Can you let us know if these address your concerns? If not, we would love to hear your feedback on how to improve the work further.

---

### Decision · Program_Chairs · 2023-01-20

**Decision:**

Accept: poster

**Justification For Why Not Higher Score:**

I am recommending an accept to this paper in spite of borderline reviews since I think the authors have addressed many points brought up in the reviews, and because the paper is interesting and adds value to the community. I am recommending a poster, since the paper provides limited novelty to a broader audience, outside of UI understanding.

**Justification For Why Not Lower Score:**

N/A

**Metareview: Summary, Strengths And Weaknesses:**

This paper presents a purely vision+language model for mobile UI understanding. Compared to past works it does not use the webpage view hierarchy and instead relies on visual cues as its input. A new pre-training method enables the authors to pre-train a large powerful model on a large new corpus. Finetuning on downstream tasks improves on previous methods in spite of them using the view hierarchy of the webpage.

Three reviewers reviewed this paper and provided two borderline reject ratings and one borderline positive rating. Their main concerns were: The novelty of the work, given that the model is similar to Flamingo, the experiments being unfair to the baselines since they do not employ pre-training, the poor presentation and missing details. The authors have made several improvements to the paper adding a lot of detail and improving the presentation of the material. I feel that the model figure can be made clearer, but apart from that, I think the presentation is much improved and this issue is resolved. In terms of novelty, the model itself is similar to Flamingo. But I disagree with the reviewers about the overall novelty of this paper beyond the model design. I think the pre-training strategy, although an extension of past work in pre-training for V+L, is still new to the mobile UI space and adds a lot of value. The pre-training dataset which will be made public adds value to the community. The model, although similar to past work, still makes small but meaningful improvements for this task. In terms of impact, I think people working in this space should read this paper and perhaps build on it. This paper may convince the community to move away from view hierarchies and use pixel information for future tasks. Finally, the reviewers complained about the baselines not having access to pre-training. I think that is the point. This paper shows that in spite of not having view hierarchies, pre-training and fine tuning with this model improves on more traditional methods. To make this argument, I think the baselines are appropriate.

Given my above arguments, I am recommending an accept in spite of the limited enthusiasm shown by the reviewers for this work.

**Note From Pc:**

if the above contains the word "oral" or "spotlight" please see: "oral" presentation means -> notable-top-5% and "spotlight" means -> notable-top-25%. As stated in our emails, we are disassociating presentation type from AC recommendations

**Summary Of Ac-Reviewer Meeting:**

Did not have AC-reviewer meetings due to a lack of response by reviewers to my request.